# Engineering Pyrrolysyl-tRNA Synthetase for the Incorporation of Non-Canonical Amino Acids with Smaller Side Chains

**DOI:** 10.3390/ijms222011194

**Published:** 2021-10-17

**Authors:** Nikolaj G. Koch, Peter Goettig, Juri Rappsilber, Nediljko Budisa

**Affiliations:** 1Institut für Chemie, Technische Universität Berlin, 10623 Berlin, Germany; nikolaj.koch@mail.tu-berlin.de; 2Institut für Biotechnologie-Bioanalytik, Technische Universität Berlin, 10623 Berlin, Germany; juri.rappsilber@tu-berlin.de; 3Structural Biology Group, Department of Biosciences, University of Salzburg, 5020 Salzburg, Austria; peter.goettig@plus.ac.at; 4Wellcome Centre for Cell Biology, University of Edinburgh, Edinburgh EH9 3BF, UK; 5Department of Chemistry, University of Manitoba, Winnipeg, MB R3T 2N2, Canada

**Keywords:** genetic code expansion, pyrrolysyl-tRNA synthetases, non-canonical amino acids, aliphatic amino acids, bioorthogonal reactive handles, azidohomoalanine, photo-methionine, protein engineering, stop codon suppression, *S*-allyl-l-cysteine

## Abstract

Site-specific incorporation of non-canonical amino acids (ncAAs) into proteins has emerged as a universal tool for systems bioengineering at the interface of chemistry, biology, and technology. The diversification of the repertoire of the genetic code has been achieved for amino acids with long and/or bulky side chains equipped with various bioorthogonal tags and useful spectral probes. Although ncAAs with relatively small side chains and similar properties are of great interest to biophysics, cell biology, and biomaterial science, they can rarely be incorporated into proteins. To address this gap, we report the engineering of PylRS variants capable of incorporating an entire library of aliphatic “small-tag” ncAAs. In particular, we performed mutational studies of a specific PylRS, designed to incorporate the shortest non-bulky ncAA (*S*-allyl-l-cysteine) possible to date and based on this knowledge incorporated aliphatic ncAA derivatives. In this way, we have not only increased the number of translationally active “small-tag” ncAAs, but also determined key residues responsible for maintaining orthogonality, while engineering the PylRS for these interesting substrates. Based on the known plasticity of PylRS toward different substrates, our approach further expands the reassignment capacities of this enzyme toward aliphatic amino acids with smaller side chains endowed with valuable functionalities.

## 1. Introduction

Research in the field of reprogrammed protein translation has now reached experimental and intellectual maturity: More than 200 non-canonical amino acids (ncAAs, i.e., a diversity that is an order of magnitude higher than that of the canonical amino acid repertoire) were introduced into proteins via various genetic code expansion routes: Selective pressure incorporation, stop codon suppression (SCS), fragment condensation, protein semisynthesis, and peptidomimetics [1]. It has been shown that AAs with non-proteinogenic functional groups can be used to manipulate, design, and elucidate protein structure, dynamics, function, allosterism, interactions, catalysis, folding, synthesis, trafficking, degradation, and aggregation [2,3,4,5,6,7,8]. 

Engineering aminoacyl-tRNA synthetase (aaRS)/tRNA pairs capable of recognizing, activating, and loading ncAAs onto their cognate tRNAs is now a well-established strategy. It enables the site-specific ribosomal incorporation of ncAAs in response to a reprogrammed codon. The most commonly used approach for this purpose is in-frame stop codon suppression, targeting the amber stop codon [9,10,11]. Hereby, the ncAA is incorporated in response to an in-frame stop codon placed at a predefined position in the protein coding sequence, ribosomally expressed either in vivo or in vitro [9,10,11]. Most aaRS variants used for SCS so far are derived from *Methanosarcina mazei/barkeri* pyrrolysyl-tRNA synthetases (*Mm*PylRS/*Mb*PylRS) or *Methanocaldococcus jannaschii* tyrosyl-tRNA synthetase (*Mj*TyrRS) [9,10,11,12]. The archaeal origin and therefore distant phylogeny is responsible for their orthogonality in bacterial and eukaryotic cells [12].

The native substrate of the PylRS is the rare proteinogenic amino acid pyrrolysine (Pyl, 1a), a lysine analog with a 4-ethyl-pyrroline-5-carboxylate ring attached to the terminal amino function of the side chain (Figure 1). The wild type enzyme can activate several Pyl variants resembling ncAAs [12]. Moreover, catalytic promiscuity is widely exploited in both native and genetically engineered classes of PylRS enzymes to enable recognition, activation, and tRNA loading of the majority of all translationally active ncAAs. [13,14,15,16]. It should be noted, however, that the majority of incorporable ncAAs with the PylRS system are characterized with flexible, long-chained, and bulky pyrrolysine analogs [13,14] or shorter but still bulky aromatic substrates, especially phenylalanine [16], tryptophan [17], and histidine [18] analogs. Therefore, a new class of PylRS enzymes capable of recognizing, activating, and tRNA loading with shorter chain ncAAs endowed with useful functional groups is of great interest. Small ncAAs with shorter side chains containing azido, thioene, fluoro, cyano, and nitroso groups can be particularly useful, e.g., for FTIR, NMR, crosslinking, and spin labeling, because longer side chains are too flexible which usually results in a loss of spectral information or the necessary proximity for specific bioorthogonal reactions [19]. Moreover, still no efficient non-canonical counterparts are available for Glu and Asp, which often form structurally important salt bridges or hydrogen-bond networks. It would be useful to modify these acidic residues, e.g., by removing their hydrogen bond donors or acceptors. Mimicking post-translational modifications of canonical amino acids (cAAs) with their genetically encoded ncAA counterparts is also an attractive application to elucidate their functions.

The main reason for the substrate promiscuity of PylRS is most likely the unique substrate binding mode with relatively nonspecific hydrophobic interactions in the large binding pocket of this enzyme. Therefore, it is not surprising that a multitude of ncAAs can be recognized and activated with very few mutations (the majority has just 2–4) mainly in the binding pocket [12]. For this reason, the PylRS is predestined for the implementation of new functions [20]. In general, an enzyme should possess two key features to ensure successful recognition of new substrates. First, the target enzyme should have low levels of the desired new activity, which in case of PylRS means enzymatic activity toward ncAAs that are highly divergent from the native substrate [21,22]. Second, sufficient stability is required to buffer destabilizing mutations necessary for active site remodeling [23,24,25]. Unfortunately, PylRS is marginally stable under standard cultivation conditions in *Escherichia coli* (*E. coli*) [26], which is also reflected by the low in vitro solubility of the enzyme [20,27]. We demonstrated that this drawback can be partly remedied with a solubility tag fused to the N-terminus of *Mb*PylRS (manuscript in preparation) which made this our enzyme of choice.

In our study, we performed mutational analyses to elucidate the structure activity relationship of a PylRS designed to incorporate *S*-allyl-l-cysteine (Sac, 1). Based on this knowledge, we engineered several new *Mb*PylRS variants on a rational and semi-rational basis, in order to incorporate a variety of small side chain ncAAs. We used an entire library of small side-chain-containing ncAAs that can be structurally and functionally categorized into five classes (Figure 1, detailed discussion in Appendix B). Briefly, they include (i) aliphatic analogs of Sac (1); (ii) bio-orthogonal tags; (iii) small ncAAs with useful spectroscopic probes; (iv) methionine analogs; and (v) substrates with a terminal alkene as site-specific chemical cleavage site (being also bio-orthogonal tags).

## 2. Results and Discussion

### 2.1. General MbPylRS and ncAA Incorporation Readout Setup

All PylRS variants used in this study are equipped with an N-terminal SmbP solubility tag [28]. As mentioned earlier, the tag restores activity by dramatically increasing the abundance of soluble and active enzyme compared to the untagged aaRS. Most likely, this phenomenon is due to an increase in kinetic stability and builds an improved and solid starting point for our enzyme engineering efforts. This fusion enzyme is used throughout this work (for sequence information see Appendix A). The Y349F mutation was included by default, as it is known to generally enhance aminoacylation and orthogonal translation systems (OTS) efficiency [13,29].

To test the efficacy of ncAAs incorporation, we used superfolder-GFP (sfGFP) as a model protein. The sfGFP-based fluorescence readout is the simplest approach for this purpose as the fluorescence intensity of intact cells is directly correlated to the amount of protein produced. The sfGFP reporter construct comprises an in-frame stop codon at position 2 (instead of an arginine triplet; sfGFP(R2 amber)). This construct has been routinely used as a readout vehicle for the in vivo suppression efficiency of the in-frame amber stop codon [30]. This reporter construct is an integral part of the OTS established in *E. coli* BL21(DE3) expression host strain. To avoid deficiency of ncAA in the cell and to detect even very low incorporation activity, we used high concentrations of ncAAs (10 mM, unless otherwise stated) in this study.

### 2.2. Testing MbSacRS for Aliphatic Substrates

We hypothesized that the previously reported SacRS could accommodate close structural aliphatic analogs of Sac (1). Therefore, this variant would be a good starting point for the evolution of SacRS toward similar small-tag substrates. To scrutinize this hypothesis, a SacRS variant (hereafter referred to as *Mb*SacRS) was created based on a codon optimized *Mb*PylRS sequence by introducing the two crucial active site mutations C313W:W382S. In addition, two previously identified advantageous N-terminal mutations T13I:I36V ([31], cf. Appendix A) were introduced as well and maintained them for all other constructs. Surprisingly, we found very low incorporation of the aliphatic substrates for the *Mb*SacRS (Appendix A). Since this specificity among very close structural analogs is relatively uncommon for both native and mutant PylRS enzymes, we set out to investigate the structure–activity relationships of the *Mb*SacRS.

### 2.3. Elucidating the Structure–Activity Relationships of MbSacRS via Rational Mutation Studies

To date, no high-resolution crystal structure of SacRS is available. Since there are only two PylRS mutations responsible for altering the substrate specificity to Sac (1), we decided to perform rational mutation studies to elucidate the role of each residue in Sac-incorporation activity. An overview of all relevant residues, whose mutations were guided by crystal structures, can be found in Figure 2. Figure 2B was especially helpful because it contains the same C313W mutation as the *Mb*SacRS.

Starting with residue S382, we reverted this position to wild-type Trp and tested less bulky amino acids. As serine was located at position 382 in the original SacRS, we investigated whether polar functional groups are necessary for Sac (1) incorporation. As shown in Figure 3A, all constructed mutants resulted in comparable Sac incorporation, with the exception of the C313W:W382F and C313W constructs. This was quite surprising since the authors in the original SacRS report found only three variants for Sac (1) incorporation and just one variant possessed the C313W mutation [33]. This finding highlights the enormous importance of quality control when constructing libraries and the need for sufficient analytics when analyzing newly found variants after screening. Figure 3A does not provide a clear picture of whether a hydroxyl group at position 382 is an advantage, since the C313W:W382A variant performs at comparable levels. In contrast, small size clearly plays an important role, with the Phe and Trp mutations being the two most inefficient variants in the group so far. Interestingly, the Trp mutant is found to have a strong cAA background incorporation. This phenomenon is perfectly in line with the literature reports. It is known that PylRS enzymes with C313W mutations and a small residue at position N311 incorporate Phe [34]. The original finding of SacRS was based on positive and negative selection rounds. The inclusion of the negative step in the selection against variants with high background incorporations clearly shows why the variant W382 was overlooked. Considering the gathered data, the W382 mutation, to smaller residues, is most likely important for restoring orthogonality of *Mb*SacRS.

We tested the four best variants as depicted in Figure 3A in a concentration-dependent manner to gain detailed information about the OTS performance in vivo. The best constructs were (C313W:W382T/Y) and showing similar activity, but twice the OTS efficiency (at 0.6 mM Sac (1)) compared to the original SacRS (Appendix A). To determine whether this was a specific property of the *Mb*PylRS, we transferred the mutations to the *Mm*PylRS, the variant resulting in the SacRS. For this variant, the equivalent of the (C313W:W382T) mutations was also observed as the best outcome, indicating a robust result (Appendix A). Interestingly the second-best variant of *Mb* and *Mm*SacRS were different. This result highlights the fact that the best mutation found in one species is not necessary the best in another, although mutations are transferable between the different PylRS systems of different species. The C313W:W382T variant was also reported to incorporate *S*-propargyl-l-cysteine (27) [35]. Therefore, we also tested the top four Sac (1) incorporating constructs also with substrate 27 and found that the (C313W:W382T/Y) constructs gave comparable results (Appendix A). All gathered data for the W382 mutations suggest that this residue only needs to be smaller than Trp to restore orthogonality. However, the efficiency of Sac incorporation differs for different residues at this position. In particular, variants with a polar OH- or SH-group at W382 position perform best. Examining the effect of the C313W mutation in our system, the data displayed in Figure 3B clearly indicates that the size of the residue at this position is the most prominent factor, albeit not the only one. No variant with a small amino acid at this position was able to incorporate Sac. By contrast, the bulkier Phe allows Sac incorporation, though at a lower level compared with all C313W mutants. Even with Met at this position, very low incorporation is detectable. The inactive C313H variant, comparable in size to C313F, suggests that a non-polar residue is necessary to be present in this microenvironment. Overall, the data collected suggest that a reduced size of the binding pocket containing the C313W mutation is essential for Sac incorporation.

### 2.4. Rationalizing Sac Incorporation Data and Creating Aliphatic Substrate Activating MbPylRS Variants

It was previously proposed that the C313W mutation is critical for activation of smaller substrates [32,33]. This hypothesis was also fully recapitulated in this work. Having thus established that the C313W mutation is critical for the incorporation of Sac (1) and probably also smaller aliphatic Sac (1) analogs, all these mutants were tested for incorporation of substrates 2, 3, 5, 7, and 10 (Figure 4). These fluorescence data showed that all variants, except the C313W mutant, did not incorporate the tested substrates. 

Unfortunately, the construct that incorporated the aliphatic ncAAs also exhibited a considerable level of cAA background incorporation. As mentioned previously, bulky C313 mutations of *Mb*PylRS lead to background Phe incorporation [20,34]. We hypothesized that the key to restoring the lost orthogonality associated with the C313W mutation would lie in the mutation of *Mb*PylRS at position N311, which is also known to be one of the two gatekeeper residues for Pyl activation. Our working hypothesis was to restore the orthogonality by increasing the size of the amino acid at this position to interfere with the Phe substrate accommodation while creating a catalytic pocket suitable for the short-chain aliphatic ncAAs. As shown in Figure 5A, the orthogonality also increases by increasing the AA size at the N311 position, until it is fully restored for the N311L:C313W variant.

An incorporation pattern of the tested aliphatic substrates was noticeable, most likely corresponding to the size of the newly created catalytic pocket. The signal intensity for the N311L:C313W variant suggests that substrate 2 is the optimal size for this pocket. This substrate is the aliphatic equivalent to Sac (1). The data supported the hypothesis that it was possible to restore orthogonality by replacing the residue at position N311 to bulkier AAs. This finding encouraged us to test all amino acids larger than valine at this position. The additional functional variants found are shown in Figure 5B. Figure 5 shows all the obtained active variants. Two of them (N311M/Q:C313W) can incorporate some of the substrates even better than the N311L:C313W construct. Interestingly, these two additional variants exhibit a different ncAA incorporation profile. The N311M construct favors substrates with carbon chain length C6, while the N311Q favors C7. The incorporation profile of the N311Q variant is similar to that of the N311L.

The key residues found here, which are important for maintaining the orthogonality of the PylRS system will facilitate future enzyme engineering efforts for the incorporation of structural analogs. The information will enable the creation of smarter PylRS-libraries; in particular, now a reduction in library size is feasible and will increase the likelihood of finding desired enzymes. This will facilitate the establishment of ready-made enzyme arsenals to test a wide variety of natural and synthetic ncAAs for translational activity. In addition, for close structural analogues, the number and type of selection cycles (e.g., negative selection) can be reduced, making the overall selection process simpler and more feasible.

### 2.5. Semi-Rational Engineering of PylRS Constructs for Small Aliphatic Substrate Incorporation

The previously described effort led to five constructs with efficient incorporation of aliphatic ncAAs. We selected two of these constructs (N311M:C313W and N311Q:C313W) for further engineering. The goal was to improve the incorporation efficiency and/or specificity of the aliphatic substrates. For this reason, we targeted residues potentially in close proximity to them. After inspecting the crystal structure (Figure 2), we planned to randomize residues A267, V366, Y349, and W382 in the active site by site-saturation mutagenesis (SSM) with NNK primers (N = A/T/C/G; K = G/T). We started with position V366 because it is located opposite to the N311 residue. Thereby we hypothesized that altering V366 might have the strongest tuning effect with respect to aliphatic ncAA recognition. The randomization of this position resulted in two new variants for each parent construct, with mutations V366A/K (Figure 6).

Figure 6A shows the OTS performance of these constructs compared to the starting N311M:C313W construct. The V366K variant did not show a strong difference in the incorporation profile compared to the starting enzyme. Interestingly, the V366A mutation resulted in a specificity shift toward long-chain aliphatic ncAAs in the incorporation profile. This phenomenon seems plausible based on the *Mm*PylRS crystal structures (Figure 2), since this mutation potentially increases the space of the binding pocket, which should facilitate the incorporation of longer substrates. Figure 6B shows that the two variants found, based on the N311Q:C313W construct, performed comparably to the parent enzyme. Similar to the N311M:C313W:V366A construct, there is also a slight shift in the incorporation profile toward longer ncAAs observable for the V366A mutant. Since this variant already favors ncAAs with C7 chain length over C6, the shift is smaller than for the N311M:C313W construct. Unfortunately, screening of randomizations A267, Y349, and W382 did not yield better performing variants. However, a variant with a markedly different incorporation profile was found (N311M:C313W:W382H). This mutant preferably incorporated the longer chained substrate 3 (Appendix A).

### 2.6. Evaluating the Incorporation of Biochemically Useful Aliphatic ncAA Analogs

We screened all generated *Mb*PylRS constructs for incorporation of the aliphatic ncAA/AA analogs listed in Figure 1 (besides 2, 3, 5, 7 and 10). Substrates 1–9, 20–26, and 28 were highly incorporated (Figure 5, Appendix A) and could be expressed in a standard *E. coli* BL21(DE) protein production strain to higher levels (up to 21 mg/L protein per culture, Table 1). Interestingly, several constructs also incorporated Sac (1) and substrate 27, with the best Sac (1) incorporating construct being N311M:C313W:V366A (Appendix A). This result clearly illustrates that multiple substrate-recognizing enzyme topologies can be based on the same scaffold. Since this variant was also able to incorporate 26, we were encouraged to perform another round of randomization to find out if its performance is enhanced or if other interesting Cys-based amino acid derivatives were incorporated. This approach led to the identification of N311M:C313W:V366A:W382N/T/Y mutants that were able to successfully incorporate substrate 28.

Substrates 11–19 with very low incorporation signals were additionally screened in release factor 1 (RF1) knock-out strains JX33, B-95.ΔA, and C321.ΔA.exp (Appendix A) [36,37,38]. Usually, strains lacking RF1 produce higher amounts of full-length protein by amber suppression. All of these RF1 knock-out strains were engineered in previous work of our group to possess the lambda DE3 lysogen encoding the T7 polymerase compatible with our reporter protein setup. Although background incorporation increased in all RF1 knockout strains (as previously observed see [33]), some setups resulted in an increased ratio of ncAA/AA incorporation compared to background incorporation levels. The best performing strain and *Mb*PylRS combinations were selected for larger scale protein production (Table 1).

### 2.7. Analytics of Canonical/Non-Canonical Amino Acids Incorporation

Larger scale protein production was performed to confirm the results from the fluorescence assays in small-scale 96-well plates. The protein yields are in good agreement with the trends observed in the fluorescence experiments. We acquired the mass spectra of the intact reporter proteins via electrospray ionization mass spectrometry (ESI-MS). To facilitate MS data evaluation for AAs with a low incorporation efficiency, a reporter protein with a C-terminal His_6_-tag was used. Table 1 shows the optimal setup of reporter protein production, the ESI-MS data, and protein yields for each substrate. The corresponding deconvoluted ESI-MS-spectra are shown in the Appendix A). The incorporation of all substrates but five (14, 17, 18, 19, and 24) was confirmed in the MS analytics. For substrate 14, the molecular weight of incorporated AA is 146.3 g/mol which is equivalent to that of glutamine (146.2 g/mol). It is known that near-cognate suppressor tRNAs, like tRNA^Gln^, read amber codons to some extent [39]. This means Gln is incorporated at amber sites when the OTS is not working and then frequently observed in MS analytics. The high reporter protein yield for a non-functioning OTS is most likely the result of the general higher background suppression observed in RF1 knock-out strains (Appendix A). Gln incorporation is also observed for the setup of substrate 16, 17. For substrates 18, 19, and 24, the molecular weight of the incorporated AA is 165.3/164.3 g/mol, which is equivalent to phenylalanine (165.2 g/mol), indicating that this PylRS leads to Phe incorporation when no or inefficient substrate is present. The observed high protein yield for the setup of substrates 18 and 19 is probably caused by a combination of the Phe incorporation activity and the use of a RF1 knock-out strain, although further analytics would be required here. The same is true for substrate 24, though on a lower level since no RF1 knock-out strain was used. Nonetheless, the fluorescence data with different *Mb*PylRS constructs clearly show that the incorporation is possible (Appendix A).

In summary, in this study, we generated 28 *Mb*PylRS variants which can incorporate 23 ncAAs and one cAA. To the best of our knowledge, 17 of these ncAAs (besides 1, 3, 4, 17, 27, and 28) were not ribosomally incorporated by amber suppression before and 20 of them were not previously incorporated with the PylRS system [33,35,40,41].

## 3. Materials and Methods

### 3.1. Canonical and Non-Canonical Amino Acids

Canonical amino acids were purchased from Carl Roth. Non-canonical amino acids were obtained from Fluorochem, Iris Biotech, Chempur, Sigma-Aldrich (Merck), Chiralix, Toronto Research Chemicals, Carl Roth, Thermo Fisher Scientific and TCI Deutschland (see Appendix A).

### 3.2. Plasmid Vector Construction

All plasmids were assembled by Golden Gate cloning and confirmed by DNA sequencing. Plasmids harboring the OTS (aaRS/tRNA^Pyl^) were constructed by cloning the target aaRS gene into the pTECH vector (Addgene plasmid #104073) [42].

### 3.3. Site-Directed and Site-Saturation Mutagenesis

Point mutations were introduced by non-overlapping inverse PCR [43]. Focused *Mb*PylRS gene libraries were also created with non-overlapping inverse PCR, but randomization was performed using mutagenic primers (with NNK, whereby N = A, T, G, or C; K = G or T) at designated positions (A267, V366, F349, and W382).

### 3.4. Analysis of SUMO-sfGFP Expression by Intact Cell Fluorescence.

For the small-scale expression of reporter constructs, *E. coli* BL21(DE) cells were used. Electrocompetent cells were transformed with the orthogonal translation system and reporter plasmids. LB agar plates for plating contained 1% glucose and corresponding antibiotics. Single colonies of clones were used for inoculation of 2 mL LB (in 14 mL tubes) with 1% glucose and appropriate antibiotics and were grown to saturation overnight. Assays were conducted in 96-well plate format. Cultures were added to each well at 1:100 dilution in ZYP-5052 auto induction medium to a final volume of 100 μL supplemented with antibiotics and ncAAs. Cells were grown in black μ-plates (Greiner Bio-One, Kremsmünster, Austria) covered with a gas permeable foil (Breathe-Easy^®^, Diversified Biotech, Dedham, MA, USA) with orbital shaking for 24 h at 37 °C. For endpoint measurements (Tecan M200, Männedorf, Switzerland), the plate foil was removed and fluorescence was measured with an 85 gain setting. For OD_600_ measurements, 50 µL of ZYP-5052 medium was pipetted into clear 96-well μ-plates and 50 µL of culture was added. Excitation and emission wavelengths for fluorescence measurements were set to 481 nm and 511 nm, respectively. Fluorescence values were normalized to the corresponding OD_600_. Biological triplicates were used for measurements of each aaRS construct. Relative fluorescence was normalized to the highest value. The data including standard deviation represent the mean of three biological replicates.

### 3.5. Library Screening

After library creation and transformation, 96 clones were picked and grown overnight in a 96-well plate in 100 µL LB with 1% glucose and appropriate antibiotics. The next day a 96-well plate with 100 µL ZYP-5052, appropriate antibiotics, and ncAAs was inoculated with 1 µL culture, grown for 24 h and measured afterwards as stated above. The 96-plate which was used for inoculation was sealed with aluminum foil and stored at 4 °C. From this plate, desired clones were analyzed via PCR gene amplification and sequencing of this PCR product afterwards. Calculations with the Toplib tool estimate the probability of finding the best performing variant to be 96% (using a yield of 85%, which is the lower limit of primer purity and therefore also the lower yield limit of created DNA constructs) [44].

### 3.6. Protein Expression

For expression of the SUMO-sfGFP variants, *E.coli* strains were used in 10 mL ZYP-5052 medium supplemented with 10 mM ncAA and appropriate antibiotics. The expression medium was inoculated with a starter culture (1:100). Shake flasks were incubated for 24 h at 37 °C while shaking at 200 rpm. Cells were harvested by centrifugation and stored at –80 °C or directly used for protein purification.

### 3.7. Protein Purification

Harvested cell pellets were resuspended (50 mM sodium phosphate, 300 mM NaCl, 20 mM imidazole, pH 8.0) and lysed with B PER^®^ Bacterial Protein Extraction Reagent (Thermo Scientific, Waltham, MA, USA) according to their protocol, with addition of phenylmethanesulfonyl fluoride (PMSF, 1 mM final concentration), DNAse, and RNAse. Cleared lysates were loaded onto a equilibrated Ni-NTA column and purified via the P-1 peristaltic pump (Pharmacia Biotech, now Cytiva, Marlborough, MA, USA ). After washing with 10 column volumes of resuspension buffer, elution buffer (50 mM sodium phosphate, 300 mM NaCl, 500 mM imidazole, pH 8.0) was applied to elute the his-tagged target proteins. The first 2 mL covering the void volume was discarded. Afterwards, the eluate (1 mL) was collected and dialyzed in cellulose film tubings against 1 L buffer (50 mM sodium phosphate, 300 mM NaCl, pH 8.0) for at least 2 h with three buffer changes. Concentrations of purified reporter proteins were determined by measuring the sfGFP chromophore absorption at 488 nm.

### 3.8. ESI-MS

Intact protein mass measurements of purified SUMO-sfGFP variants were performed by electrospray LC-MS on a Waters H-class instrument with a Waters Acquity UPLC protein BEH C4 column (300 Å, 1.7 μm, 2.1 mm × 50 mm). The following gradient used a flow rate of 0.3 mL/min: A: 0.01% formic acid in H_2_O; B: 0.01% formic acid in MeCN. 5–95% B 0–6 min. Mass analysis was conducted with a Waters Xevo G2-XS QTof analyzer. Proteins were ionized in positive ion mode applying a cone voltage of 40 kV. Raw data were analyzed employing the maximum entropy deconvolution algorithm. The data were exported and plotted with QtiPlot (version 0.9.9.7).

## 4. Conclusions

We were motivated by previous works implying that the PylRS, naturally specialized for large bulky substrates, could be redesigned for small substrates. Therefore, we elucidated the structure–activity relationship of a specific *Mb*PylRS (*Mb*SacRS), designed to incorporate the shortest non-bulky ncAA (*S*-allyl-l-Cysteine, 1) possible to date. Based on this knowledge, we have designed *Mb*PylRSs for the incorporation of aliphatic amino acids and various derivatives thereof, which are useful in biochemistry and structural biology. Some of the incorporated substrates (allylglycine, 8 and propargylglycine, 13) were recently synthesized in vivo in *E.coli* and would, therefore, open up the possibility of coupling metabolic engineering and ncAA incorporation [45]. This approach could eliminate the need to add ncAAs to the cultivation medium, which would drastically decrease costs and simplify associated applications.

In addition, valuable information was gathered about the role of specific residues in the active site that are responsible for the orthogonality of PylRS. We were able to determine key residues that maintain orthogonality upon engineering the *Mb*PylRS enzyme to accommodate smaller substrates. This achievement will facilitate future enzyme engineering efforts for the incorporation of structural analogs. To this end, PylRS libraries can now be created with a smaller size, which simultaneously increases the likelihood of finding the desired enzymes and reduces the amount of work required to do so by eliminating negative selection.

## Figures and Tables

**Figure 1 ijms-22-11194-f001:**
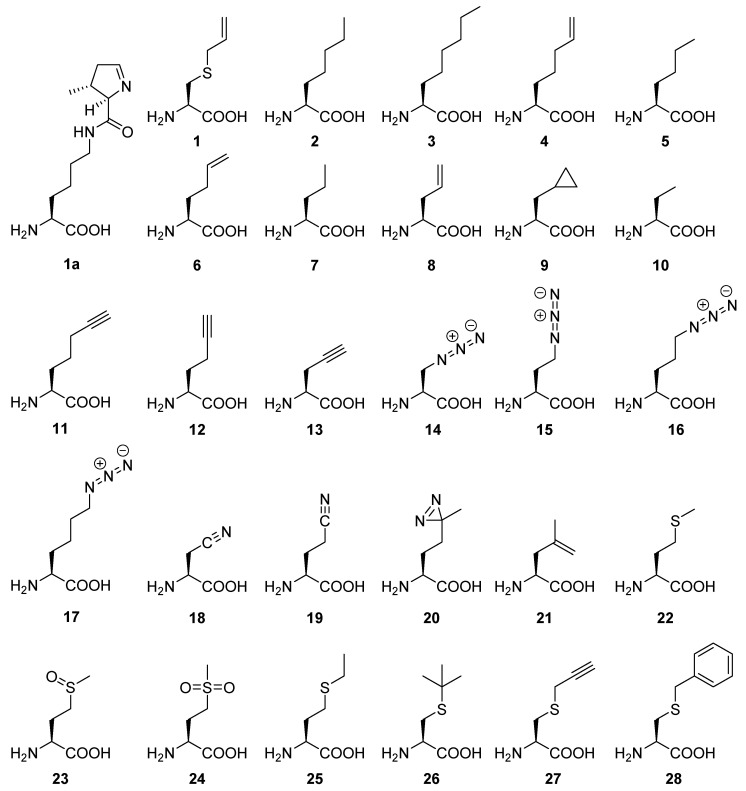
Survey of amino acids used in this study. Chemical structure of pyrrolysine (1a), *S*-allyl-l-cysteine (1), (*S*)-2-aminoheptanoic acid (2), (*S*)-2-aminooctanoic acid (3), (*S*)-2-aminohept-6-enoic acid (4), (*S*)-2-aminohexanoic acid (5), (*S*)-2-aminohex-5-enoic acid (6), (*S*)-2-aminopentanoic acid (7), (*S*)-2-aminopent-4-enoic acid (8), (*S*)-2-amino-3-cyclopropylpropanoic acid (9), (*S*)-2-aminobutyric acid (10), (*S*)-2-aminohept-6-ynoic acid (11), (*S*)-2-aminohex-5-ynoic acid (12), (*S*)-2-aminopent-4-ynoic acid (13), (*S*)-2-amino-3-azidopropanoic acid (14), (*S*)-2-amino-4-azidobutanoic acid (15), (*S*)-2-amino-5-azidopentanoic acid (16), (*S*)-2-amino-6-azidohexanoic acid (17), (*S*)-2-amino-3-cyanopropanoic acid (18), (*S*)-2-amino-4-cyanobutanoic acid (19), (*S*)-2-amino-5,5′-azi-hexanoic acid (20), (*S*)-2-amino-4-methylpent-4-enoic acid (21), l-methionine (22), l-methionine sulfoxide (23), l-methionine sulfone (24), l-ethionine (25), *S*-tert-butyl-l-cysteine (26), *S*-propargyl-l-cysteine (27), *S*-benzyl-l-cysteine (28).

**Figure 2 ijms-22-11194-f002:**
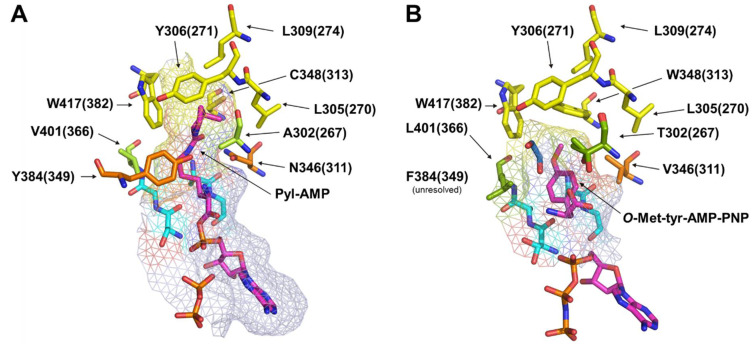
Microenvironment of the active sites derived from the crystal structures of *Mm*PylRS and the *Mm*OmeRS mutant. These structures guided the rational mutation approach. Shown are critical residues forming the active site. Since only structures of *M. mazei* are available, these were used in a homology model for *M. barkeri*. Residue numbers in brackets reflect the numbering of *M. barkeri,* while numbers not in brackets refer to *M. mazei*. (**A**) Wild-type *Mm*PylRS (PDB ID: 2Q7H) [20] with bound Pyl-AMP. (**B**) Mutant *Mm*OmeRS (PDB ID: 3QTC) [32] with bound *O*-Methyl-tyrosine-AMP-PNP (*O*-Met-tyr-AMP-PNP).

**Figure 3 ijms-22-11194-f003:**
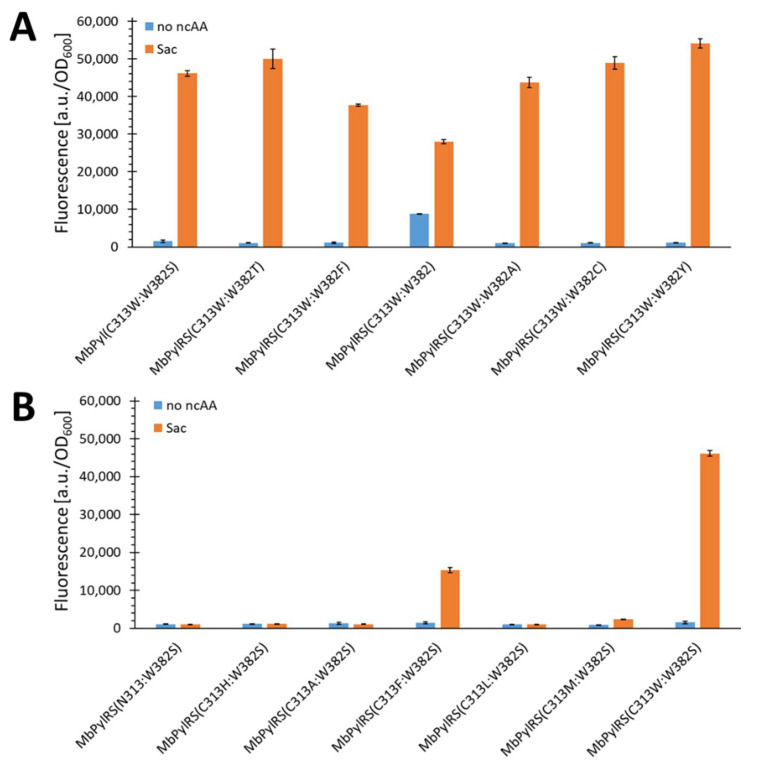
Comparison of Sac incorporation efficiency for *Mb*PylRS constructs (**A**) *Mb*PylRS(C313W) and variants mutated at position W382 and (**B**) *Mb*PylRS(W382S) with variants mutated at position C313. The fluorescence was measured for intact *E. coli* BL21(DE3) cells expressing the SUMO-sfGFP(R2amber) reporter protein. The data (incl. standard deviation) represent the mean of three biological replicates.

**Figure 4 ijms-22-11194-f004:**
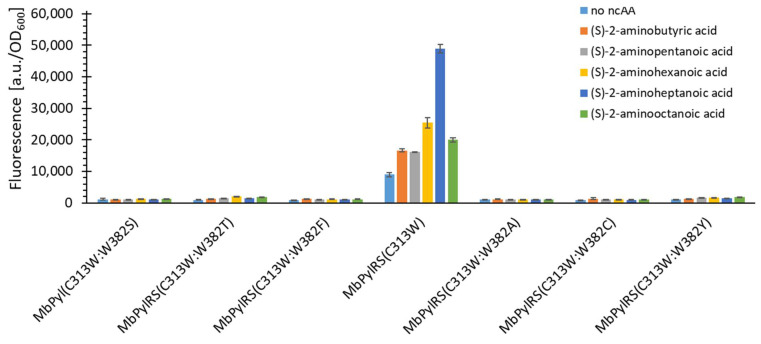
Comparison of the efficiency of aliphatic ncAA incorporation for *Mb*PylRS(C313W) constructs mutated at position W382. The fluorescence was measured for intact *E. coli* BL21(DE3) cells producing the SUMO-sfGFP(R2amber) reporter protein. The data (incl. standard deviation) represent the mean of three biological replicates.

**Figure 5 ijms-22-11194-f005:**
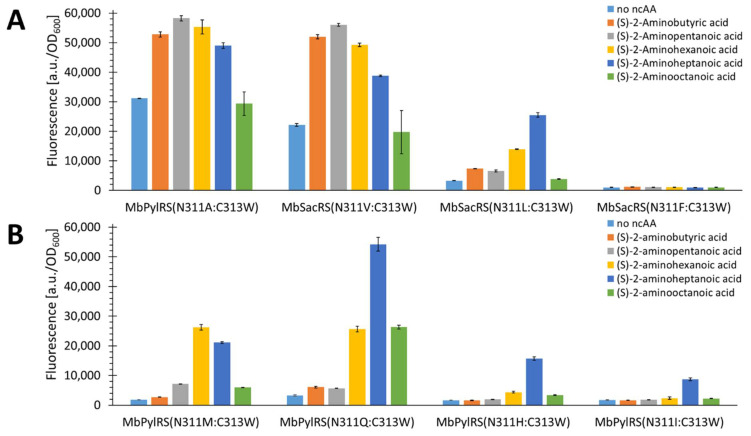
Comparison of aliphatic ncAA incorporation efficiency for *Mb*PylRS(C313W) constructs mutated at position N311. (**A**) Construct N311A with increased binding pocket size and additional mutations that stepwise decrease the size of the binding pocket. (**B**) Four additional active variants, found after screening of all 19 (all cAAs besides glycine) possible constructs. The fluorescence was measured for intact *E. coli* BL21(DE3) cells producing the SUMO-sfGFP(R2amber) reporter protein. The data (incl. standard deviation) represent the mean of three biological replicates.

**Figure 6 ijms-22-11194-f006:**
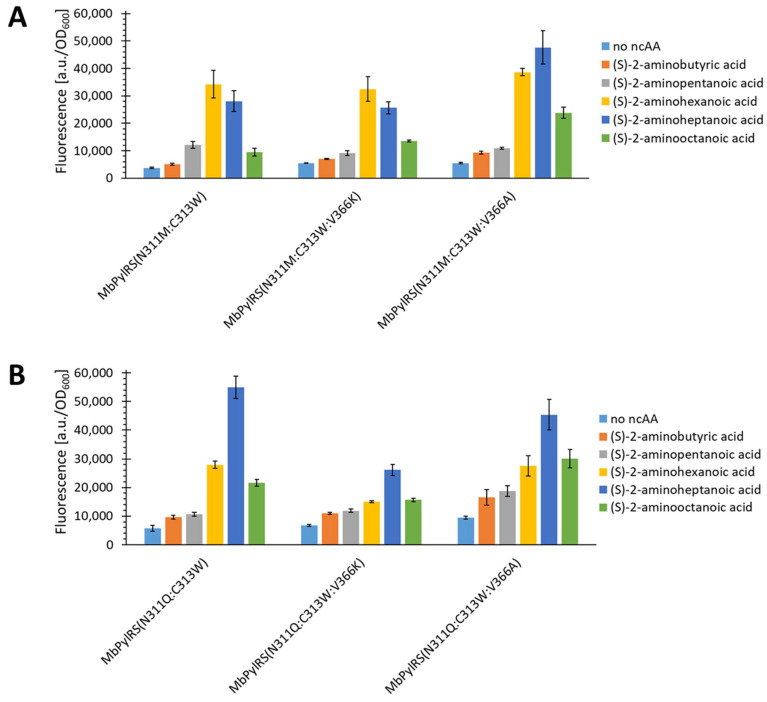
Comparison of aliphatic ncAA incorporation efficiency for (**A**) *Mb*PylRS(N311M:C313W) and (**B**) *Mb*PylRS(N311Q:C313W) constructs both mutated at position V366. The fluorescence was measured for intact *E. coli* BL21(DE3) cells producing the SUMO-sfGFP(R2amber) reporter protein. The data (incl. standard deviation) represent the mean of three biological replicates.

**Table 1 ijms-22-11194-t001:** Optimal reporter protein production setup, calculated and observed molecular weights of the reporter proteins His_6_-SUMO-sfGFP(R2AA)-strep (a) /SUMO-sfGFP(R2AA)-His_6_ (b) and protein production yields per liter of culture. The masses were determined by ESI-MS of intact proteins.

AA	*E. coli* Strains ^1^	*Mb*PylRS Construct	Reporter Construct	Calculated Mass [Da]	Observed Mass [Da]	Δ Mass [Da]	Protein Yield [mg∙L^−1^] ^2^
**1**	BL21	N311M:C313W:V366A	a	40,194.9	40,196	1.1	10.8
**2**	BL21	N311Q:C313W	a	40,178.8	40,180	1.2	5.1
**3**	BL21	N311M:C313W:V366A	a	40,192.9	40,194	1.1	1.6
**4**	BL21	N311M:C313W:V366A	a	40,176.8	40,179	2.2	1.7
**5**	BL21	N311M:C313W	a	40,164.8	40,166	1.2	1.9
**6**	BL21	N311M:C313W:V366K	a	40,162.8	40,164	1.2	1.4
**7**	BL21	N311M:C313W	a	40,150.8	40,153	2.2	1.2
**8**	BL21	N311M:C313W	a	40,148.8	40,150	1.2	0.7
**9**	BL21	N311M:C313W	a	40,162.8	40,164	1.2	1.4
**10**	BL21	N311M:C313W:V366A	a	40,136.8	40,195	58.2	0.8
**11**	C321.ΔA.exp	N311M:C313W	b	38,990.9	38,992	1.1	5.1
**12**	C321.ΔA.exp	N311M:C313W	b	38,976.9	38,979	2.1	4.9
**13**	C321.ΔA.exp	N311M:C313W	b	38,962.8	38,965	2.2	11.3
**14**	JX33	N311M:C313W	b	38,979.8	38,996	16.2	4.3
**15**	C321.ΔA.exp	N311M:C313W	b	38,993.8	38,994	0.2	14.2
**16**	C321.ΔA.exp	N311M:C313W	b ^4^	39,007.9	39,007	0.9	5.3
**17**	C321.ΔA.exp	N311M:C313W	b	39,021.9	38,997	24.9	6.4
**18**	C321.ΔA.exp	N311Q:C313W:V366K	b	38,963.8	39,015	51.2	19
**19**	C321.ΔA.exp	N311Q:C313W:V366K	b	38,977.8	39,014	36.2	19.9
**20**	BL21	N311M:C313W:V366K	a	40,190.8	40,194	3.2	4.8
**21**	BL21	N311M:C313W	b	38,978.9	38,982	3.1	2.6
**22**	BL21	N311M:C313W	b	38,998.9	38,998	0.9	3.6
**23**	BL21	N311Q:C313W	b	39,014.9	39,012	2.9	4.6
**24**	BL21	N311Q:C313W:V366K	b	39,030.9	39,015	15.9	10.7
**25**	BL21	N311Q:C313W	b	39,013	39,014	1	9.8
**26**	BL21	N311M:C313W:V366A	a	40,210.9	40,211	0.1	21
**27**	BL21	N311M:C313W:V366A	a ^3^	-	-	-	-
**28**	BL21	N311M:C313W:V366A: W382N	b	39,061	39,063	2	1.1

^1^ all DE3, ^2^ yield per liter of cell culture, ^3^ was not purified, ^4^ not the main peak

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
