# Peer review of "Engineering Pyrrolysyl-tRNA Synthetase for the Incorporation of Non-Canonical Amino Acids with Smaller Side Chains"

_ijms, 2021, doi:10.3390/ijms222011194_

Round 1

Reviewer 1 Report

In this manuscript, Koch et al. explore PylRS mutants that are capable of incorporating small aliphatic non-canonical amino acids (ncAAs). Using non-sense suppression of GFP and measuring fluorescence in E. coli, they mutagenize six residues in PylRS (A267, V366, N311, C313, Y349 and W382) to achieve incorporation of a variety of small side chain ncAA. The approach is very similar to one used previously by these authors to engineer a PylRS variant that incorporates S-allylcysteine (Exner et al. 2016; ChemBioChem). The variant identified in this previous paper was used as the starting place for these studies. Here, the authors identify mutations that decrease non-specific incorporation and improve the incorporation of various ncAA. The variants identified here will be useful for incorporating the ncAAs highlighted in this study and engineering further PylRS variants that incorporate other useful ncAA. This study will be of interest to those involved in genetic code expansion. Although the data is interesting, I have some points that I believe must be addressed:

Major Points

  1. Throughout the paper, data in each figure is plotted as relative fluorescence which makes it difficult to compare between figures. The authors should plot normalized fluorescence units (a.u./OD600) as is done in some of the supplemental figures to make the data more comparable.
  2. In the figures, data is presented as averages (it is never mentioned what each bar represents) with error bars. The authors should indicate how many replicates each bar represents, as well as what the errors bars represent in the figure legends. Additionally, the authors could plot individual replicate data points on top of the bars which would give the reader a better sense of the data.
  3. Statistical comparisons between variants should be performed to support conclusions made in the text. For example, on line 185 the authors state “all constructed mutants led to a comparable Sac incorporation”. It would help to know which variants are actually statistically different from each other.
  4. The authors should include a control with cells not expressing the PylRS/SacRS variant to establish what the baseline read-through is. It is not clear if the ~ 3% relative fluorescence that occurs with most variants comes from the synthetase or if that is background for coli/the system.
  5. On line 201, the authors state the best construct is C313W:W328T, but looking at the data in Figure 2A it seems C313:W328Y is the best. It possible that upon statistical comparison these two variants are identical, but the authors should clarify this discrepancy in the text.
  6. Figure S3 shows SacRS variants with mutations at position 379 but on line 202 in the manuscript where the figure is reference, the authors say the C313W:W328T mutant has similar activity with double the OTS efficiency at lower Sac concentrations but the C313W:W328T variant is not shown in that figure. Maybe I mis-understood, but the data does not seem to support this conclusion.
  7. On line 220, the authors say that C313F enables Sac incorporation, but at lower levels than C313W. The authors should include the C313W variant in this figure to enable comparison. Currently, it is impossible to verify this claim as C313W is only shown in part A of this figure and all the fluorescence values are reported relatively.

Minor Points

  1. There are reference errors on lines 218, 257 and 292
  2. It would significantly improve the paper if the authors included a figure showing the PylRS active site and where the mutations they are making are located relative to the binding pocket.

Author Response

Reviewer 1: In this manuscript, Koch et al. explore PylRS mutants that are capable of incorporating small aliphatic non-canonical amino acids (ncAAs). Using non-sense suppression of GFP and measuring fluorescence in E. coli, they mutagenize six residues in PylRS (A267, V366, N311, C313, Y349 and W382) to achieve incorporation of a variety of small side chain ncAA. The approach is very similar to one used previously by these authors to engineer a PylRS variant that incorporates S-allylcysteine (Exner et al. 2016; ChemBioChem). The variant identified in this previous paper was used as the starting place for these studies. Here, the authors identify mutations that decrease non-specific incorporation and improve the incorporation of various ncAA. The variants identified here will be useful for incorporating the ncAAs highlighted in this study and engineering further PylRS variants that incorporate other useful ncAA. This study will be of interest to those involved in genetic code expansion. Although the data is interesting, I have some points that I believe must be addressed:

Reviewer 1 Question 1 Throughout the paper, data in each figure is plotted as relative fluorescence which makes it difficult to compare between figures. The authors should plot normalized fluorescence units (a.u./OD600) as is done in some of the supplemental figures to make the data more comparable.

Answer Reviewer 1 Question 1): We thank the reviewer for bringing this issue to our attention. Now, all corresponding figures display normalized fluorescence with units (a.u./OD600).

Reviewer 1 Question 2): In the figures, data is presented as averages (it is never mentioned what each bar represents) with error bars. The authors should indicate how many replicates each bar represents, as well as what the errors bars represent in the figure legends. Additionally, the authors could plot individual replicate data points on top of the bars which would give the reader a better sense of the data.

Answer Reviewer 1 Question 2): We thank the reviewer for this suggestion. For clarity, we have described this in detail in the methods section. But now also included this in the figure caption. Since we performed triplicate experiments, and the standard deviations were generally small, we see no need to report the 3 data points for each column. We would also refrain from doing so for clarity reasons, to avoid overcrowding the figures.

Reviewer 1 Question 3): Statistical comparisons between variants should be performed to support conclusions made in the text. For example, on line 185 the authors state “all constructed mutants led to a comparable Sac incorporation”. It would help to know which variants are actually statistically different from each other.

Answer Reviewer 1 Question 3 As mentioned in the text, this statement refers to Figure 3A. The Figure has been adapted, now the incorporation efficiencies are clearly shown, and the reader can easily verify this statement. They do not perform exactly the same, but they are in the same range. We have clarified and specified that in the text.

Since only triplicates were used it is not clear to us what kind of statistics would be applicable for a sample set of 21 (3x7) total data points.

Reviewer 1 Question 4): The authors should include a control with cells not expressing the PylRS/SacRS variant to establish what the baseline read-through is. It is not clear if the ~ 3% relative fluorescence that occurs with most variants comes from the synthetase or if that is background for coli/the system.

Answer Reviewer 1 Question 4): We thank the reviewer for this highly relevant suggestion. We have already performed this experiment earlier with an almost identical setup (See: H. Tseng et al. Molecules 2020, 25, 4418, doi:10.3390/molecules25194418.) Therefore, we believe that it is not necessary to repeat it in this study as it provides little to no additional information There is, some readthrough in E. coli, as is in any other organism (but, only to a very small extent). In addition, there is an increase in read-through with PylRS an no ncAAs but this is very low as well. Since we performed ESI-MS with all ncAAs/PylRS mutant combinations, we could have detected a substantial amount of cAA incorporation, as we did for substrates 10, 14, 17, 18, 19 and 24.

Reviewer 1 Question 5):  On line 201, the authors state the best construct is C313W:W328T, but looking at the data in Figure 2A it seems C313:W328Y is the best. It possible that upon statistical comparison these two variants are identical, but the authors should clarify this discrepancy in the text.

Answer Reviewer 1 Question 5): We thank the reviewer for this critical remark. Indeed, in figure 2A (now Figure 3A) construct C313:W328Y is the best. However, as mentioned in the text, since this is measured at very high concentrations, we measured the four best performing variants in a concentration dependent manner to elucidate which variant also performs better even at lower concentrations (Figure S3). We found that C313:W328T performed slightly better than C313:W328Y. Yet, the margin of error is very small, thus we have changed this in the main text.

Reviewer 1 Question 6): Figure S3 shows SacRS variants with mutations at position 379 but on line 202 in the manuscript where the figure is reference, the authors say the C313W:W328T mutant has similar activity with double the OTS efficiency at lower Sac concentrations but the C313W:W328T variant is not shown in that figure. Maybe I mis-understood, but the data does not seem to support this conclusion.

Answer Reviewer 1 Question 6): We thank the reviewer who brought this labeling error to our attention. The constructs are clearly MbSacRS and MbSacRS(S382T/Y/C).

Figure S3 clearly shows that the MbSacRS construct at 0.6 mM ncAA has a fluorescence value of 19000±200 and MbSacRS(S382T) of 40000±2000. Thus, within the standard deviation MbSacRS(S382T) is twice as efficient.

Reviewer 1 Question 7): On line 220, the authors say that C313F enables Sac incorporation, but at lower levels than C313W. The authors should include the C313W variant in this figure to enable comparison. Currently, it is impossible to verify this claim as C313W is only shown in part A of this figure and all the fluorescence values are reported relatively.

Answer Reviewer 1 Question 7): This issue was addressed with formatting the Y-axis with [a.u./OD600], now they can be compared.

Reviewer 1 Question 8 – Minor points): There are reference errors on lines 218, 257 and 292

Answer Reviewer 1 Question 8 – Minor points): These errors were corrected in the revised Ms

Reviewer 1 Question 9 – Minor points): It would significantly improve the paper if the authors included a figure showing the PylRS active site and where the mutations they are making are located relative to the binding pocket.

Answer Reviewer 1 Question 9 – Minor points): According to this suggestion we prepared the new Figure 2 in the revised manuscript.

Reviewer 2 Report

In the present manuscript, Nikolaj G. Koch and coworkers have engineered the Pyrrolysyl-tRNA synthetase (PylRS) from Methanosarcina barkeri (Mb) so that to be able to incorporate non-canonical amino acids (ncAAs) into proteins. They have performed a mutational analysis of the PylRS to elucidate the structure-activity relationship dedicated to ncAAs incorporation and succeeded in incorporating ncAAs of small side chains, all derived from the previously reported S-allyl-L-cystein (Sac). They thus have created several new MbPyRS variants able to selectively incorporate ncAAs, which makes it possible to significantly increase the diversity of the physicochemical properties of amino acids within recombinant proteins. This achievement will undoubtedly have multiple and important applications. However, the extent of the work carried out is, as it stands, not fully appreciable. The manuscript would therefore benefit from some improvements in order to render its reading more confortable.

Some parts of manuscript are long and wordy. It concerns mainly the introduction and part 2.1 (section "Results and Discussion") that takes the reader away from the concrete objectives of the work. An effort of concision should be made. The authors should also pay attention not to overuse abbreviations. For instance GCE (Genetic code expansion) is never re-used in the text. Likewise, sentences such as "Structure activity relationship" should not be abbreviated.

The authors should clarify how the experiments are normalized with respect to each other (quantification of cells expressing the PylRS variant in each experiments, ...)

A SacRS (PylRS derivatives that incorporate the Sac ncAA) has been previously published (ChemBiohem communication 2017). It is made from the Methanosarcina mazei PylRS. The reason why the authors have chosen to use Methanosarcina barkeri PylRS it the present study is unclear. The importance of residues corresponding to C313 and W382 was already established in the previous study. Their re-investigation in the present study (Figure 2) appears thus redundant.

For additional minor comments:

Colors for supplementary figures S1 and S2 should be the same as within main figures; otherwise it is misleading (e.g; no ncAA is in blue in the main figures and oranges in Supp. Figures S1 and S2)

In the text; the "Error Reference Source not found" has to be fixed (two occurrences).

Author Response

Reviewer 2: In the present manuscript, Nikolaj G. Koch and coworkers have engineered the Pyrrolysyl-tRNA synthetase (PylRS) from Methanosarcina barkeri (Mb) so that to be able to incorporate non-canonical amino acids (ncAAs) into proteins. They have performed a mutational analysis of the PylRS to elucidate the structure-activity relationship dedicated to ncAAs incorporation and succeeded in incorporating ncAAs of small side chains, all derived from the previously reported S-allyl-L-cystein (Sac). They thus have created several new MbPyRS variants able to selectively incorporate ncAAs, which makes it possible to significantly increase the diversity of the physicochemical properties of amino acids within recombinant proteins. This achievement will undoubtedly have multiple and important applications. However, the extent of the work carried out is, as it stands, not fully appreciable. The manuscript would therefore benefit from some improvements in order to render its reading more confortable.

Reviewer 2 Question 1): Some parts of manuscript are long and wordy. It concerns mainly the introduction and part 2.1 (section "Results and Discussion") that takes the reader away from the concrete objectives of the work. An effort of concision should be made.

Answer Reviewer 2 Question 1): We thank the reviewer for this suggestion; in our revised version we have streamlined the manuscript. In particular, the proposed sections were shortened and restructured, to present the information in a more concise manner.

Reviewer 2 Question 2: The authors should also pay attention not to overuse abbreviations. For instance, GCE (Genetic code expansion) is never re-used in the text. Likewise, sentences such as "Structure activity relationship" should not be abbreviated.

Answer Reviewer 2 Question 2: GCE as a single abbreviation has been deleted. However, the abbreviation “SAR” is quite common and widespread and widely used in the relevant literature. It was nevertheless changed.

Reviewer 2 Question 3: The authors should clarify how the experiments are normalized with respect to each other (quantification of cells expressing the PylRS variant in each experiments, ...)

Answer Reviewer 2 Question 3: We thank the reviewer for bringing this issue to our attention. Since reviewer 1 demanded adjustment of the Y-axes with normalized fluorescence values, this point is properly addressed.

Reviewer 2 Question 4 A SacRS (PylRS derivatives that incorporate the Sac ncAA) has been previously published (ChemBiohem communication 2017). It is made from the Methanosarcina mazei PylRS. The reason why the authors have chosen to use Methanosarcina barkeri PylRS it the present study is unclear.

Answer Reviewer 2 Question 4: We appreciate this relevant comment and agree this issue was not thoroughly explained in the original manuscript This point was addressed by rewriting the sentence in line 82/83 in the introduction. The information about the required properties of an enzyme should be clarifying and is provided immediately before our rationale for choosing MbPylRS. Briefly, we observed a substantial activity increase in MbPylRS mutants with a fused SmbP tag. Although additional mutations further reduce stability, the tagged mutants still have an advantage. These results are in preparation for another publication.

Reviewer 2 Question 5: The importance of residues corresponding to C313 and W382 was already established in the previous study. Their re-investigation in the present study (Figure 2) appears thus redundant.

Answer Reviewer 2 Question 5: We thank to the referee for bringing this to our attention. In the aforementioned study, the mutations were merely found but not investigated for their role. For example, it was unclear, whether the serine in W382S is really needed, or whether only a small amino acid was sufficient. To this end, we performed necessary analyses in our study, demonstrating that the W382S mutation is not actually necessary for the incorporation of Sac (Figure 3) but instead establishes a stronger orthogonality. This was the crucial hint, which encouraged the idea to mutate N311 and keep W382 unmutated (Figure 5).

Reviewer 2 Question 6 – Minor points): Colors for supplementary figures S1 and S2 should be the same as within main figures; otherwise, it is misleading (e.g; no ncAA is in blue in the main figures and oranges in Supp. Figures S1 and S2)

Answer Reviewer 2 Question 6 – Minor points): The colors were adapted as suggested.

Reviewer 2 Question 7 – Minor points): In the text; the "Error Reference Source not found" has to be fixed (two occurrences)..

Answer Reviewer 2 Question 7 – Minor points): This problem was fixed.

Round 2

Reviewer 1 Report

The authors have done a good job at addressing both reviewer comments. After reading through the manuscript again, I believe the data supports the conclusions. I commend the authors on the edits, especially adding more rationale and hypotheses to motivate the experiments. It makes the manuscript much more logical to read. I still identified a few minor points listed below to improve clarity of the manuscript. Of note, the authors should check that the figures referenced in each part of the text match the data they are describing.

Minor Points

  1. In the abstract, the newly added text on line 17 is not clear. I believe there is a typo but was not exactly sure the point the authors were trying to make.
  2. In figure 2, the authors should label the Pyl-AMP and O-Methyl-tyrosine-AMP-PNP.
  3. On line 164, that authors have added that “the W328S mutation is most likely important for restoring orthogonality” but based on the data it looks like a wide range of mutations at W328 restore orthogonality. It might be more correct to state “mutation of W328 is most likely important for restoring orthogonality” rather than highlighting just the serine mutation.
  4. On line 171, the authors say they picked the four best variants shown in Figure 3A. I don’t believe this is the data shown in Figure 3. Figure 3 also only have one panel and is referred to as an A and B part on lines 171 and 183, respectively. I think the data that is being referred to is Figure 2A/B. Currently, as it stands, there is no reference to Figure 2B in the manuscript.
  5. The flow of the paragraph on page 6 makes it difficult to read. The authors start off by talking about mutations at position 328, then talk about mutations at position 313 and finish with a conclusion about position 328. For flow, it might be better to move the discussion of mutations at 313 to its own paragraph after the discussion of mutations at position 328 is finished.
  6. I believe the reference to figure 4 on line 199 should be figure 3.
  7. Figure 4B is not referenced/discussed in the text. I think the reference to Figure 5 on line 225 should be to Figure 4B.
  8. References to figure 5 are not correct. On line 250, I believe the reference to Figure 6 should be to Figure 5. On line 256, the reference to Figure 6A should be to 5A. On line 262, the reference to Figure 6B should be to 5B.

Author Response

(Reviewer 1)

The authors have done a good job at addressing both reviewer comments. After reading through the manuscript again, I believe the data supports the conclusions. I commend the authors on the edits, especially adding more rationale and hypotheses to motivate the experiments. It makes the manuscript much more logical to read. I still identified a few minor points listed below to improve clarity of the manuscript. Of note, the authors should check that the figures referenced in each part of the text match the data they are describing.

Minor Points

  1. In the abstract, the newly added text on line 17 is not clear. I believe there is a typo but was not exactly sure the point the authors were trying to make.

Answer: The “the” was in the wrong position. I hope it is correct now.

  1. In figure 2, the authors should label the Pyl-AMP and O-Methyl-tyrosine-AMP-PNP.

Answer: Has been changed

  1. On line 164, that authors have added that “the W328S mutation is most likely important for restoring orthogonality” but based on the data it looks like a wide range of mutations at W328 restore orthogonality. It might be more correct to state “mutation of W328 is most likely important for restoring orthogonality” rather than highlighting just the serine mutation.

Answer: We rephrased it to more general statement.

  1. On line 171, the authors say they picked the four best variants shown in Figure 3A. I don’t believe this is the data shown in Figure 3. Figure 3 also only have one panel and is referred to as an A and B part on lines 171 and 183, respectively. I think the data that is being referred to is Figure 2A/B. Currently, as it stands, there is no reference to Figure 2B in the manuscript.

Answer: The Figure numbering was wrong because there were two Figure 2. Now it is rightfully referenced as Figure 3A. Figure 3B is mentioned in line 187.

  1. The flow of the paragraph on page 6 makes it difficult to read. The authors start off by talking about mutations at position 328, then talk about mutations at position 313 and finish with a conclusion about position 328. For flow, it might be better to move the discussion of mutations at 313 to its own paragraph after the discussion of mutations at position 328 is finished.

Answer: We thank the reviewer for this remark. We have moved that part to line 182 and modified the sentence slightly

  1. I believe the reference to figure 4 on line 199 should be figure 3.

Answer: As stated above, the Figure numbering was wrong and is now correct.

  1. Figure 4B is not referenced/discussed in the text. I think the reference to Figure 5 on line 225 should be to Figure 4B.

Answer: We thank the reviewer for this observation. An additional sentence, line 225-226, has been added for clarification.

  1. References to figure 5 are not correct. On line 250, I believe the reference to Figure 6 should be to Figure 5. On line 256, the reference to Figure 6A should be to 5A. On line 262, the reference to Figure 6B should be to 5B

Answer: As stated above, the Figure numbering was wrong and is now correct.

Reviewer 2 Report

All the concerns previously raised have been properly addressed.

Author Response

All the concerns previously raised have been properly addressed.

Answer: We thank Reviewer 2 for the constructive contributions.